



# A new estimate of oceanic CO₂ fluxes by machine learning reveals the impact of CO₂ trends in different methods

Jiye Zeng[1], Tsuneo Matsunaga[1], and Tomoko Shirai[1]

[1]National Institute for Environmental Studies, Tsukuba, 305-8506, Japan

*Correspondence to*: Jiye Zeng (zeng@nies.go.jp)

**Abstract.** Global oceans have absorbed a substantial portion of the anthropogenic carbon dioxide (CO₂) emitted into the atmosphere. Data-based machine learning (DML) estimates for the oceanic CO₂ sink have become an import part of the Global Carbon Budget in recent years. Although DML models are considered objective as they impose very few subjective conditions in optimizing model parameters, they face the challenge of data scarcity problem when applied to mapping ocean CO₂

concentrations, from which air-sea CO₂ fluxes can be computed. Data scarcity forces DML models to pool multiple years' data for model training. When the time span extends to a few decades, the result could be largely affected by how ocean CO₂ trends are obtained. This study extracted the trends using a new method and reconstructed monthly surface ocean CO₂ concentrations and air-sea fluxes in 1980-2020 with a spatial resolution of 1x1 degree. Comparing with six other products, our results show a smaller oceanic sink and the sink in early and late year of the modelled period could be overestimated if ocean

CO₂ trends were not well processed by models. We estimated that the oceanic sink has increased from $1.79\pm0.47$ PgC yr$^{-1}$ in 1980s to $2.58\pm0.20$ PgC yr$^{-1}$ in 2010s with a mean acceleration of 0.027 PgC yr$^{-2}$.

## 1 Introduction

The carbon dioxide (CO₂) emitted into the atmosphere by human activities has been considered a key factor causing abnormal climate changes for decades and the oceans play a crucial role in mitigating against the increase of atmospheric CO₂ (Sabine

et al., 2004; Khatiwala et al., 2013; McKinley et al., 2016). The Global Carbon Budget 2021 (Friedlingstein et al., 2021) derived the oceanic sink from the average of flux estimates by biogeochemistry models and by data-based machine learning (DML) models. Although DML models are more objective in terms of optimization as they do not impose subjective conditions on the valid ranges of model parameters, under learning or overfitting due to data scarcity could result in false extrapolations to unsampled domains. Reconstructing surface ocean CO₂ (simply refer to as ocean CO₂ or simply CO₂ thereafter) by DML

models for flux estimate also faces the same challenge.

The Surface Ocean CO₂ Atlas (SOCAT) Database Version 2021 (Sabine et al., 2013; Pfeil et al., 2013; Bakker et al.,2016) is a combined product of in situ measurements of ocean CO₂ by internationally coordinated efforts. It shows a composite sampling map covering most areas of the oceans. However, only a small portion of the oceans had samples in any single year





and the sampling is extremely unbalance in season and geography (refer to Supplement 1). This situation forced DML models to pool multiple years data for training. The dilemma is while ocean $CO_2$ tends to track the increase in atmospheric $CO_2$ closely (Fay and McKinley, 2013; Bates et al., 2014), the large seasonal and special viabilities of $CO_2$ up to a few hundred μatm makes it difficult to detect its annual increase rate in the order of a few μatm per year. Current methods for solving the problem include normalizing ocean $CO_2$ to a reference year (Takahashi et al. 2009; Sasse et al., 2012 & 2013; Nakaoka et al., 2013;

Zeng et al., 2014), including a linear time-dependent term in regression (Fay and McKinley, 2013; Iida et al., 2015, 2021; Jones et al., 2015; Watson et al., 2020), and using atmospheric $CO_2$ as a predictor to make DML models learn the trend implicitly (Landschützer et al., 2016; Gregor and Gruber, 2021; Denvil-Sommer et al., 2019; Chau et al., 2021). The former two methods assume a constant trend for the whole model period. It could be a good approximation when the time span is short but tends to overestimate the annual increase rate of $CO_2$ in early years and underestimate the rate in later years with the

time extended to a few decades. As for the third approach, Landschützer et al. (2016) has shown that if several years of observations were withheld from the training at the beginning or end, then the trend is ill constrained during these years, leading to larger errors. The evaluation of Gloege et al. (2021) also shows that the method was less capable in reconstructing variability at decadal timescales.

There are two camps of DML models in ocean $CO_2$ reconstruction in terms of data pooling strategy. One camp treated the global oceans as one entity (Takahashi et al., 2009; Sasse et al., 2013; Nakaoka et al., 2013; Zeng et al., 2014; Denvil-Sommer et al., 2019; Chau et al., 2021). Methods in this camp sacrifices the accuracy of region-dependent trends to ease the scarcity problem. An apparent global trend was thus derived and used. Another camp divided the oceans into clusters with similar biogeochemical properties. Sasse et al. (2012, 2013) and Landschützer et al. (2013,2016) are early pioneers in this camp. They

used Self-Organization Map (SOM) for clustering in the first step and then used different regression methods in the second step for making prediction. The two-step method of Landschützer et al. (2016) was also used by Laruelle et al. (2017), Watson et al., 2020, and Gloege et al. (2021). Other clustering methods include geographical blocking (Iida et al., 2015; Watson et al., 2020), K-mean clustering (Gregor et al., 2019; Gregor and Gruber, 2021), and $CO_2$ biome clustering (McKinley et al., 2011; Fay and McKinley, 2013; Gregor et al., 2019; Watson et al., 2020). Post smoothing is necessary for clustering methods to

make the spatial changes of $CO_2$ contingent. An advantage of the two-step method is that the clustering brings together regions with similar seasonality and similar co-variability with predictors. While this approach could partially reduce factors that obstruct the extraction of ocean $CO_2$ trend, the data scarcity problem tends to become more severe in some clusters.

This study used three DML models to extend the work of Zeng et al. (2014) by using a new method to extract ocean $CO_2$

trends in decadal scales. It largely reduced the error of using a constant trend to reconstruct ocean CO2 that resulted in overestimates of $CO_2$ uptake in the starting and ending years. Our models yielded smaller air-sea $CO_2$ fluxes comparing to six other products included in Global Carbon Budget 2021 (Friedlingstein et al., 2021).





## 2 Method

### 2.1 Model Setup

Following Zeng et al. (2014, 2017), we express the nonlinear dependence of ocean $CO_2$ on time and biogeochemical variables as

$$CO2W = f(SST, dSST, SSS, CHL, MLD, LAT, LON) + f(year) \qquad (1)$$

where SST stands for sea surface temperature, SSS for sea surface salinity, CHA for chlorophyll-a concentration, MLD for mixed layer depth, LAT for latitude and LON for longitude. The sine and cosine converted values of LON were used to make

the circular variable contingent. We replaced the month variable of Zeng et al. (2014) with SST anomaly (dSST) against the annual mean to harmonize the seasons of the two hemispheres. The function of year represents the increase rate of $CO_2$, which was a constant in Zeng et al. (2014). We deployed three databased machine learning (DML) methods to model the rate: Random Forest (RF), Gradient Boost Machine (GBM), and Feedforward Neural Network (FNN). Using multiple models has the merit points that we can check overfitting better and compensate their weakness with each other.


RF was proved to be a robust method for modelling carbon flux at global scale (Zeng et al., 2020) and was applied to global ocean $CO_2$ mapping recently (Gregor et al., 2019). We used the python library of Ranger (Weight & Ziegler, 2017) that implements the regression algorithm using two-stage randomization procedure to partition trees. A tree is assigned a subset of the training data randomly sampled with repetition; then it is recursively split into binary nodes until the number of data points

in terminal nodes becomes no larger than a specified number. In each split, the RF randomly selects a subset of predictor variables and searches them for splitting points that minimize node impurity (Ishwaran, 2015). In making a prediction, a set of predictors are passed through branches of nodes according to the splitting rule until the journey ends up in a terminal node. The mean of the target variable in the terminal node is taken as an estimate. Then the average estimate of all terminal nodes is used as the prediction. One can see that predictions are confined in the range of observations. Sensitive factors for configuring

the RF include the number of trees and the number of data points in terminal nodes (Zeng et al., 2020). The default setting includes 500 trees and 5 data points. We raised the data points to 100 based on out experiments with ocean $CO_2$ data. The configuration yielded good validation results.

GBM is also a decision-tree based machine learning model and emerged in the ocean $CO_2$ mapping recently (Gregor et al.,

2019; Gregor and Gruber, 2021). Like RF, GBM combines weak learners into a single strong learner (Natekin and Knoll, 2013), but in an iterative fashion. Trees are added one at a time, and existing trees in the model are not changed. The gradient descent procedure is used to optimize parameters of new trees to reduce the loss of predictions made with old trees. As the tree parameters are used in making predictions, the target could be extrapolated beyond the range of observations. We used the python library of LightGBM (Ke et al., 2017). By experimenting with ocean $CO_2$ data and using the RF as a reference, we



found that LightGBM performs well with 500 trees, minimum of 100 terminal nodes in a tree, and 100 data points in a terminal
node.

FNN has been used for ocean CO2 mapping since early 2010s (e.g., Landschützer et al., 2013; Zeng et al., 2014). FNN has a
layered structure, including an input layer, one or more hidden layers, and an output layer. Neurons between adjacent layers
are fully connected. Details of FNN can be found in Svozil et al. (1997) and abundant of other references. We used python's
MLPRegressor with one hidden layer and 256 hidden neurons. The early stop of training iteration was set to 300. From
experiments with the ocean $CO_2$ data, we found that the FNN was sensitive to those parameters when LAT and LON were
included as predictors. The configuration yielded results well harmonized with those of the RF and GBM. In contrast to the
RF and GBM, observations of the target variable are not used for making predictions. While this benefits data mining for
making new discoveries, overfitting and extrapolation are more likely to emerge.

## 2.2 Data

We extracted monthly $CO_2$ fugacity (fCO2) from the track-gridded database of the Surface Ocean $CO_2$ Atlas (SOCAT) version
2021 (Sabine et al., 2013; Pfeil et al., 2013; Bakker et al., 2016). We relaxed the criteria of Zeng et al. (2014) for data selection:
(i) fCO2 values smaller than 700 μatm and larger than 100 μatm and (ii) salinity larger than 15.0. A total of 273,434 data points
were extracted from 1x1 degree grids for 1980-2020. We confine fCO2 training data set to be post-1980 due to large
uncertainties in early measuring techniques (Sasse et al., 2013). The sources of predictor variables are shown in Table 1. The
values of CHL and MDL were scaled by log(1+CHL) and log(1+MDL) to reduce the skewness of sample distribution. We
used the monthly surface air pressure (Ps) of the fifth generation ECMWF atmospheric reanalysis of the global climate (ERA5)
to convert reconstructed fCO2 to partial pressure for flux calculation. The partial pressure of air $CO_2$ (pCO2A) came from
NOAA's Marine Boundary Layer Reference (Conway et al., 1994; Dlugokencky et al., 2019) and the monthly wind speed
(WIND) from ERA5.

For training models, raw ocean $CO_2$ data were normalized to the reference year 2000 by

$$CO2W^{norm}(year) = CO2W^{raw}(year) - \sum_{i=2000}^{year} sign(i - 2000) \cdot rate(year). \qquad (2)$$

$CO_2$ concentrations were reconstructed by adding the rate correction of Eq.2 to the predicted $CO_2$. The fugacity was converted
to partial pressure by the method of Weiss (1974). $CO_2$ fluxes were calculated by the difference of $CO_2$ partial pressures
between air (pCO2A) and sea (pCO2W):

$$flux = k_w k_0 (pCO2W - pCO2A), \qquad (3)$$

where $k_w$ is the gas transfer velocity, and $k_0$ the solubility of $CO_2$ in seawater. The two parameters were obtained based on
Wanninkhof (2014).



### 2.3 Rate Extraction

We used iteration method of Zeng et al. (2014) with variable length of data to estimate the annual increate rate of ocean $CO_2$ at decadal scale. The method fits the dependence of $CO_2$ on year (the second term in Eq.1) by linear regression, subtracts the trend from observations, and then used the RF, GBM, and FNN to model the nonlinear relationship between the residual and

predictors (the first term in Eq.1). The shortest data length is three years: data of the target year plus one year before and after the year. The longest data length is 41 years for the target year 2000, i.e., all data in 1980-2020 were included. These results provide valuable information on how to determine the annual rates.

### 3. Results

### 3.1 Annual Rate

The annual increase rates of ocean $CO_2$ obtained using the iteration method and the longest available data are shown in Fig. 1a along with the global annual increase rates of air $CO_2$ from the Global Monitoring Laboratory of NOAA (Conway et al., 1994) and the moving average of the rate with window size of 11 years (5 years before and after a given year). Ideally the rate for a target year should be extracted with the shortest data length possible. Because of data scarcity and the complexity of $CO_2$ dependence on time and the biogeochemical properties of seawaters, the rates fluctuate dramatically when data length is short

and converges with increasing data length (Fig. 1b-d). Sutton et al. (2019) concluded that the number of years of observations needed (YON) to detect a statistically significant trend over variability range from 8 to 15 years at several open ocean sites. It is reasonable to assume that YON for open oceans would not be smaller. Because it is difficult to determine the smallest stabilization length, and the rates do not change much after 10 to 15 years, we used the rates with the maximum data length. As a result, the rate for 2000 (1.71 $\mu$atm yr$^{-1}$) is equivalent to the long-term trend of 1980-2020.


The rates appear to track the monotonical increase of the decadal mean rates of air $CO_2$ in 1997-2015 and remain flat with small changes in 1990-1996 during which air $CO_2$ rates experienced a long period of decline. Fig. 1b-d reveal that it requires at least 7 years' observations to detect the trend at the global scale. Comparing to a fixed site, the spatial variability imposes an additional interference to trend detection. The YON was about 20 years in early 1990s (Fig. 1b) when samples were few

and incomplete in season (Supplement Fig. 1) and about 9 years in early 2010s (Fig. 1d) when samples were relatively abundant. Around year 2000, the trend became rather stable when the data length exceeds 10 years (Fig. 1c) but continued to go up and down with data length. One of the reasons must be that the data points after 2000 overwhelm data before the year. With the extension of data length, the trend tends to be affected more and more by data in later years.

The orange line in Fig. 1a represents the annual rates for data normalization in this study. We assumed that the rates before 1990 is the same as in 1990, i.e., 1.46 $\mu$atm yr$^{-1}$. This value is close to the rate of 1.50 $\mu$atm yr$^{-1}$ used by Takahashi et al. (2009)



and Zeng et al. (2014), and to the mean decadal trend of air $CO_2$ (1.49 ppm yr$^{-1}$) in the same period. Further, we assume that the rate ratio between ocean and air $CO_2$ after 2015 is the same as that in 2015 (0.941) and used the ratio to calculate the annual rate of ocean $CO_2$ in 2016-2020.

## 3.2 Model Performance

We used a so-called leave-one-year-out (LOYO) method to validate model performance. Giving N years of data, N validations were done by setting aside one year's data for validation and using other N-1 years of data for training. A model's performance was evaluated by biases in all validation years. The method has an advantage over the conventional n-fold validation method in that the validation data of LOYO are more likely come from unsampled domains of the training data.

As a test for the method itself, we used it with raw $CO_2$ data to investigate the long-term trend. In the first round, the residue of prediction minus observation shows a significant ($R^2$=0.971) negative correlation with time (Supplement Fig. 2a). This is expected as the $CO_2$ concentrations in early years tend to be overestimated by the models trained with increased $CO_2$ in later years and vis versa; therefore, the negative slope (-1.352 µatm yr$^{-1}$) can be considered as an approximation of the increase trend. In the second around, we applied the LOYO method to the normalized $CO_2$ by the trend. The residue gives a regression slope of -0.208 µatm yr$^{-1}$ with $R^2$=0.495 (Supplement Fig. 2b). The process was repeated with data normalized by the sum of previous trends. After three rounds, the slope became negligible (Supplement Fig. 2c). Eventually we obtained a trend of 1.593 µatm yr$^{-1}$. It is smaller than the trends around 2000 obtained by the iterative regression method since unbalanced sampling makes it impossible for the LOYO method to eliminate overestimate/underestimate by itself.

We applied the LOYO method to the data normalized by the rates in Fig. 1a. A small trend (0.079 µatm yr$^{-1}$) exists in the biases (Supplement Fig. 3). This trend was added to the rates in Fig. 1a to form the final rates in Table 2 for data normalization. The performance of the three DML models were evaluated by the LOYO method with normalized $CO_2$. The correlation coefficient $R^2$ between observations and model predictions is 0.718 for the RF, 0.713 for the GBM, and 0.681 for the FNN; and all three models yielded negligible mean biases (Supplement Fig. 4). In term of training time, the GBM is faster than the RF and the two are much faster than the FNN. A short training time is critical for diagnose analysis and for such a case as rate extraction when many iterations are required for many years. In addition, the RF appeared to be most stable and resistant to overfitting. When the number of data points was much larger than the number of trees, the RF would produce a good result; and when many trees were used, adjusting the number of trees had little effect on the result. The GBM was also resistant to overfitting but less stable than the RF. The FNN was most sensitive to overfitting and took much longer time to train than the RF and GBM. The FNN is also more difficult to configure as the number of hidden layers and neurons, and the number of training iterations could affect the result significantly depending on the number of predictors and data points.





We compared the trends of the biases using LOYO with data normalized by the rates in Table 2 and the rate obtained by LOYO
above. The results (Fig. 2a) indicate that using a constant rate tends to underestimate $CO_2$ concentrations in those years that
are further apart from the reference year. This would be translated to larger air-sea CO2 fluxes in those years than they would
be.  The trend became insignificant when the rates in Table 2 were used to normalize $CO_2$ data (Fig. 2b).

### 3.2 Flux

Figure 3 shows the annual fluxes of our models and the spatial distribution of the mean fluxes in 1980-2020. The mean fluxes
of the biogeochemical models in Global Carbon Budget 2001 (GCB-2021) (Friedlingstein et al., 2021) is also shown in the
figure as a reference. The uncertainties of NIES-ML3 (assemble mean of the three models) were calculated by multiplying the
mean biases of the three models (Table 2) with the flux change per unit $CO_2$ change, which is about 0.18 PgC yr$^{-1}$ µatm$^{-1}$ in
our study. The flux change with time of NIES-ML3 is parallel to that of GCB-2021. The mean flux of NIES-L3 in 1980-2020
is 1.39 PgC yr$^{-1}$, which is smaller than the mean flux of GCB_2021 by 0.81 PgC yr$^{-1}$ in the same period. The difference can be
attributed to riverine inputs and the sink in coastal areas. The former was estimated to be 0.45±0.18 GtC yr$^{-1}$ by Jacobson et
al. (2007) and 0.78±0.41 PgC yr$^{-1}$ by Resplandy et al. (2018). The product of Landschützer et al. (2020) includes $CO_2$ estimate
in the coastal areas. Using their data, we estimated that the costal sink missed by our product is about 0.11 GtC yr$^{-1}$. By adding
this sink and the average riverine input of 0.61 GtC yr$^{-1}$ to the fluxes of NIES-ML3, we estimated that the ocean sink has
increase from 1.79±0.47 PgC yr$^{-1}$ in 1980s to 2.58±0.20 PgC yr$^{-1}$ in 2010s with a mean acceleration of 0.027 PgC yr$^{-2}$.


The spatial distribution patterns of the mean annual flux of NIES-ML3 in 1980-2020 (Fig. 4) agree well with those from  GCB-
2021 in 2011-2020. In term of flux per unit area, most northern oceans above 30°N appeared to be strong sinks, especially the
northern Atlantic. However, strong winter convection and upwelling made the Aleutian Basin a large source. Areas around the
tropical zone were mostly strong sources. A large portion of the southern oceans in the 30°S-60°S band absorb atmospheric at
the rate of 10 to 20 gC m$^{-2}$ yr$^{-1}$. Further south, sources and sinks intertwine from region to region. In view of the zonal sum of
fluxes, the tropical areas were the net emitter (-0.63 PgC yr$^{-1}$) with a peak (-0.44 PgC yr$^{-1}$) at about 4.5°S (Fig. 5). The northern
and southern oceans absorbed about 0.93 PgC yr$^{-1}$ and 1.08 PgC yr$^{-1}$ respectively. The North and South temperate zones played
as the most important sinks.

### 3.3 Comparisons

We compared NIES-ML3 with six DML models (Table 3) included GCB-2021. The comparisons are relative, i.e., the fluxes
in Fig. 6a-f are not the global sum of the products under comparison but the sum for those grids where both products have data.
As each products used different parameters for flux calculation and the choice of different wind products could affect the flux
largely (Roobaert et al., 2018), our comparisons focus on the relative changes, not the absolute differences.

The comparison with NIES-NN reals the error potentially existed in products that included a time-dependent linear term in regression or used a constant trend for data normalization. The flux differences are small in 1991-2006, but much larger in early and late years (Fig. 6a). NIES-NN used a constant rate of 1.54 µatm yr$^{-1}$ to normalize data to the reference year set to 2000. Fig. 2a indicate that the differences can be attributed to the underestimates of ocean $CO_2$. The JMA-MLR product also shows an arch-shaped flux trend (Fig. 6b) as NIES-NN does. Again, the differences are larger in the beginning and ending

years than in others. This is expected as the regression method of JMA-MLR includes a linear term of time, which is equivalent to using a constant trend for data normalization.

Instead of using explicit trends to normalize data or including linear term of time in regression, MPI-SOMFFN, CMEMS-FFNN, CSIR-ML6, and OceanSODA-ETHZ used atmospheric CO2 as a predictor so that their regression models could learn

the trend implicitly. The day of year is also a predictor of CSIR-ML6. Their comparisons with NIES-ML3 present different patterns (Fig. 6c-f). Except for OceanSODA-ETHZ, the flux differences tend to be smaller in early years and increase with time after 2000. The arch-shaped trend can be detected visually in MPI-SOMFFN, CMEMS-FFNN, and CSIR-ML6 after 1993. The regression models of MPI-SOMFFN, CMEMS-FFNN, and NIES-NN are all FNN. This indicates a FNN can only extract the "correct trend" partially from air $CO_2$. It is interesting to note that their fluxes around 2000 become close to that of NIES-

MLD. The trend of OceanSODA-ETHZ resemble that of NIES-ML3 the most. While OceanSODA-ETHZ and CSIR-ML6 included FNN and GBM (gradient boot machine), the former shows much larger fluxes before 1990. These flux trend patterns could be attributed partly to different clustering and ensemble schemes. JMA-MLR divided the oceans into geographically connected regions; MPI-SOMFFN used a self-organization map to classify data into 16 clusters; CMENS-FFNN is consisted of an ensemble of FNN regressions for randomly selected data in each month, which can be considered as clustering in time;

CSIR-ML6 and OceanSODA-ETHZ are consisted of a series of clusters and corresponding regressions of different models.

## 4. Summary

We have extracted the annual increase rates of ocean $CO_2$ at decadal scales with three types of machine learning models. The rates appear to track the decadal mean rates of atmospheric $CO_2$ after 1995. Data scarcity made it difficult to estimate the rates before 1990, but assuming a constant rate proves to be a good approximation as the atmospheric rates show no significant

trend. Using the LOYO validation methods with normalized ocean $CO_2$ data reveals that, in contrast to a constant rate, the time dependent rates significantly reduced the biases of model predictions, especially in the beginning and ending years of the prediction period.

The global annual air-seas fluxes obtained using the reconstructed ocean $CO_2$ concentrations exhibit a similar long-term trend to that of the mean fluxes of the biogeochemical models of GCB-2021. By adding the sink (0.11 GtC yr$^{-1}$.) of coastal areas



missed by this study and the riverine input of 0.61 GtC yr$^{-1}$ to the fluxes of this study, we estimated that the ocean sink has increase from 1.79±0.47 PgC yr$^{-1}$ in 1980s to 2.58±0.20 PgC yr$^{-1}$ in 2010s with a mean acceleration of 0.027 PgC yr$^{-2}$.

The comparisons with six other products included in GCB-2021 indicates that using a constant rate for data normalization or including a linear time-dependent term in regression would overestimate fluxes in the early and late years of the model period, especially when the time spans to a few decades. Embedding implicit rate in regression by using atmospheric $CO_2$ as a predictor could reduce the bias, but the degree of improvement depends largely on model configuration and data clustering. As our rates can only represent the apparent behaviours of the global oceans at the best and are subjected to uncertainties due to different degrees of data scarcity in each year, further studies from different angles are needed to improve the accuracy of ocean $CO_2$
flux estimate.

**Author contribution**

Jiye Zeng: Model experiment design, data processing, and draft layout. Tsunao Matsunaga: Advice on satellite data. Tomoko Shirai: Result checking and advice on carbon budget issues.

**Data and Code Availability**

The product of reconstructed ocean CO2 fugacity and air-sea fluxes can be accessed at https://db.cger.nies.go.jp/DL/10.17595/20220311.001.html.en under data doi:10.17595/20220311.001 (Zeng, 2022). The supplement material includes the python code for model initialization. As the code was written particularly to work with our data structure, we prefer to provide the complete source upon request so that we can give clear advice case by case.

**Acknowledgements**

The Surface Ocean CO2 Atlas (SOCAT) is an international effort, endorsed by the International Ocean Carbon Coordination Project (IOCCP), the Surface Ocean Lower Atmosphere Study (SOLAS) and the Integrated Marine Biosphere Research (IMBeR) program, to deliver a uniformly quality-controlled surface ocean CO2 database. The many researchers and funding agencies responsible for the collection of data and quality control are thanked for their contributions to SOCAT.

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





Table 1. Data sources. CO2W: ocean $CO_2$ fugacity. pCO2A: mole fraction ratio of atmospheric $CO_2$. CHL: chlorophyll-a concentration; SST: sea surface temperature. SSS: sea surface salinity. MLD: mix layer depth. WIND: wind speed. Ps: surface pressure.

| Variable | Units | *Resolutions* | Source URL | DOI or Version | Reference |
|---|---|---|---|---|---|
| CO2W | µatm | Monthly, 1×1 degree. | https://www.socat.info/ | Version-2021 | Sabine et al., 2013; Pfeil et al., 2013; Bakker et al., 2016. |
| pCO2A | ppm | Monthly, 0.05 sine latitude. | https://www.esrl.noaa.gov/gmd/ccgg/mbl/ | | Conway et al., 1994 and Dlugokencky et al., 2019 |
| CHL | mg m⁻³ | Monthly climatology, 0.083x0.083 degree. | https://oceancolor.gsfc.nasa.gov/cgi/l3 | 10.5067/AQUA/MODIS/L3M/CHL/2018 | Hu et al., 2012. |
| SST | °C | Monthly, 1×1 degree. | https://psl.noaa.gov/data/gridded/data.cobe.html http://ds.data.jma.go.jp/tcc/tcc/library/MRCS_SV12/explanation/cobe_sst_e.htm | | Ishii et al., 2005. |
| SSS | g g⁻³ | Monthly climatology, 1×1 degree. | https://www.nodc.noaa.gov/OC5/woa18/woa18data.html | | Zweng et a., 2019 |
| MLD | m | Monthly climatology, 1×1 degree. | https://www.nodc.noaa.gov/OC5/woa18/woa18data.html | | |
| WIND | m s⁻¹ | Monthly, 0.25×0.25 degree. | https://cds.climate.copernicus.eu/cdsapp#!/dataset/reanalysis-era5-single-levels-monthly-means?tab=form | 10.24381/CDS.F17050D7 | |
| Ps | m s⁻¹ | Monthly, 0.25×0.25 degree. | Same as WIND | Same as WIND | |






Table 2. Annual rates and LOYO validation results. Rate: rates used for data normalization and $CO_2$ reconstruction. Bias: prediction minus observation. STD: standard deviation of the biases. $R^2$: correlation coefficient between prediction and observation. ND: number of data points.

| Year | Rate | RF | | | GBM | | | FNN | | | ND |
|---|---|---|---|---|---|---|---|---|---|---|---|
| | ($\mu$atm yr$^{-1}$) | Bias ($\mu$atm) | STD ($\mu$atm) | $R^2$ | Bias ($\mu$atm) | STD ($\mu$atm) | $R^2$ | Bias ($\mu$atm) | STD ($\mu$atm) | $R^2$ | |
| 1980 | 1.538 | -3.459 | 11.397 | 0.895 | -3.087 | 12.092 | 0.885 | -5.776 | 13.386 | 0.843 | 339 |
| 1981 | 1.538 | 1.344 | 11.708 | 0.879 | 1.425 | 12.615 | 0.860 | 0.328 | 11.963 | 0.875 | 610 |
| 1982 | 1.538 | 1.998 | 15.532 | 0.551 | 2.268 | 16.528 | 0.491 | 3.789 | 18.017 | 0.380 | 441 |
| 1983 | 1.538 | -2.270 | 16.289 | 0.573 | -2.158 | 16.049 | 0.586 | -2.601 | 19.640 | 0.381 | 273 |
| 1984 | 1.538 | 0.650 | 18.233 | 0.629 | 0.108 | 18.117 | 0.634 | -1.274 | 19.106 | 0.592 | 504 |
| 1985 | 1.538 | 3.560 | 15.113 | 0.586 | 3.672 | 15.846 | 0.545 | 2.801 | 14.897 | 0.605 | 708 |
| 1986 | 1.538 | 1.586 | 16.624 | 0.703 | 2.406 | 17.779 | 0.657 | 1.408 | 14.836 | 0.763 | 897 |
| 1987 | 1.538 | 1.824 | 15.941 | 0.504 | 1.019 | 16.138 | 0.496 | 0.815 | 17.260 | 0.425 | 1587 |
| 1988 | 1.538 | -4.016 | 13.229 | 0.823 | -3.035 | 13.772 | 0.816 | -4.924 | 14.930 | 0.771 | 1010 |
| 1989 | 1.538 | -3.950 | 27.739 | 0.538 | -2.384 | 29.272 | 0.492 | -6.740 | 31.099 | 0.404 | 1117 |
| 1990 | 1.538 | -3.613 | 12.049 | 0.837 | -3.602 | 12.243 | 0.832 | -3.557 | 12.343 | 0.830 | 889 |
| 1991 | 1.553 | 0.490 | 12.588 | 0.811 | 0.365 | 13.004 | 0.799 | -0.625 | 13.885 | 0.770 | 2011 |
| 1992 | 1.539 | 3.834 | 15.136 | 0.620 | 3.623 | 15.951 | 0.583 | 3.498 | 17.087 | 0.526 | 2521 |
| 1993 | 1.576 | 1.145 | 15.751 | 0.737 | 1.035 | 16.128 | 0.725 | 1.761 | 16.953 | 0.694 | 3398 |
| 1994 | 1.605 | -3.205 | 20.712 | 0.731 | -2.384 | 21.430 | 0.715 | -4.036 | 22.130 | 0.690 | 3981 |
| 1995 | 1.577 | -1.780 | 19.298 | 0.652 | -1.800 | 19.243 | 0.654 | -0.480 | 19.812 | 0.636 | 6157 |
| 1996 | 1.599 | -1.459 | 20.418 | 0.729 | -1.011 | 20.605 | 0.725 | -2.446 | 21.740 | 0.691 | 6091 |
| 1997 | 1.613 | 2.503 | 22.817 | 0.734 | 2.733 | 23.337 | 0.721 | 2.707 | 24.357 | 0.697 | 4335 |
| 1998 | 1.698 | 1.683 | 21.860 | 0.556 | 1.976 | 22.959 | 0.510 | 2.907 | 23.859 | 0.467 | 5860 |
| 1999 | 1.749 | -2.124 | 21.757 | 0.693 | -1.425 | 22.678 | 0.669 | -3.816 | 23.640 | 0.632 | 4081 |
| 2000 | 1.794 | -3.090 | 26.342 | 0.599 | -2.533 | 27.194 | 0.575 | -1.595 | 27.883 | 0.556 | 4656 |
| 2001 | 1.804 | -4.300 | 23.231 | 0.705 | -3.778 | 22.703 | 0.720 | -4.469 | 24.230 | 0.679 | 4855 |
| 2002 | 1.808 | 0.600 | 18.504 | 0.674 | 0.540 | 18.315 | 0.680 | 0.407 | 19.884 | 0.623 | 6760 |
| 2003 | 1.813 | -0.481 | 18.796 | 0.660 | -0.688 | 19.522 | 0.633 | -2.296 | 20.726 | 0.582 | 7001 |
| 2004 | 1.822 | 0.481 | 16.927 | 0.746 | 0.664 | 17.338 | 0.733 | 1.103 | 18.128 | 0.707 | 8077 |
| 2005 | 1.848 | -1.180 | 16.819 | 0.760 | -1.077 | 17.015 | 0.755 | 0.156 | 18.415 | 0.714 | 9575 |
| 2006 | 1.861 | 0.478 | 18.753 | 0.765 | 0.547 | 18.767 | 0.765 | 0.873 | 20.518 | 0.718 | 12192 |
| 2007 | 1.861 | -0.132 | 20.361 | 0.688 | -0.472 | 20.152 | 0.695 | -0.302 | 22.100 | 0.633 | 12326 |
| 2008 | 1.905 | 0.308 | 20.295 | 0.741 | 0.473 | 20.552 | 0.734 | 0.220 | 21.776 | 0.701 | 11524 |
| 2009 | 1.930 | 2.498 | 20.484 | 0.691 | 2.093 | 20.557 | 0.690 | 3.551 | 22.125 | 0.635 | 11571 |
| 2010 | 1.942 | 0.717 | 18.485 | 0.720 | 0.539 | 18.761 | 0.712 | 0.449 | 19.973 | 0.674 | 12822 |
| 2011 | 2.022 | 1.477 | 22.692 | 0.681 | 1.642 | 22.803 | 0.677 | 3.442 | 23.392 | 0.655 | 13353 |
| 2012 | 2.082 | -0.502 | 22.875 | 0.682 | -0.726 | 22.952 | 0.680 | -0.675 | 23.389 | 0.667 | 13393 |
| 2013 | 2.159 | -0.697 | 21.486 | 0.695 | -0.725 | 22.075 | 0.678 | -0.238 | 22.101 | 0.678 | 11366 |
| 2014 | 2.283 | 3.069 | 23.902 | 0.682 | 2.469 | 23.288 | 0.700 | 2.539 | 25.135 | 0.651 | 13445 |
| 2015 | 2.357 | -1.954 | 22.905 | 0.708 | -2.217 | 23.017 | 0.705 | -2.025 | 24.323 | 0.671 | 13153 |
| 2016 | 2.395 | -0.634 | 17.929 | 0.773 | -0.647 | 18.117 | 0.768 | 0.185 | 18.945 | 0.747 | 14915 |
| 2017 | 2.420 | 0.163 | 19.695 | 0.769 | 0.005 | 19.700 | 0.769 | 1.794 | 21.663 | 0.719 | 14693 |
| 2018 | 2.422 | 0.634 | 20.126 | 0.704 | 0.373 | 20.387 | 0.697 | 0.430 | 22.574 | 0.628 | 12098 |
| 2019 | 2.455 | -1.513 | 20.551 | 0.727 | -1.628 | 20.265 | 0.734 | -2.131 | 22.488 | 0.672 | 11416 |
| 2020 | 2.431 | -0.354 | 22.937 | 0.719 | -0.056 | 22.848 | 0.721 | -0.560 | 24.389 | 0.682 | 7433 |






Table 3. Datasets for comparison. Machine learning models include random forest (RF), gradient boost machine (GBM), feedforward neural network (FNN), Support vector machine for regression (SVR) and multiple linear regression (MLR). Predictors include atmospheric $CO_2$ (COA), ocean $CO_2$ climatology (CO2C), sea surface temperature (SST), sea surface temperature anomaly (dSST), sea surface salinity (SSS), sea surface height (SSH), chlorophyll-a (CHL), chlorophyll-a anomaly (dCHL), mixed layer depth (MLD), wind speed (WIND), time (TIME), latitude (LAT), and latitude (LON).

| Dataset | Period | Reference | Regression | Trend | Clustering | Predictors |
|---|---|---|---|---|---|---|
| NIES-ML3 | 1980-2020 | This study | RL, GBM, and FNN. | Year dependent rates | 1 global cluster. CO2 normalized to 2000 using variable annual rates | SST, dSST, SSS, CHL, MLD, LAT, LON. |
| NIES-NN | 1980-2020 | Zeng et al. (2014). (doi:10.17595/20210806.001) | FNN | A linear trend | 1 global cluster. CO2 normalized to 2000 using rate of 1.54 μatm yr-1. | SST, SSS, CHL, MLD, dSST. |
| JMA-MLR | 1990-2020 | Iida et al. (2021) https://www.data.jma.go.jp/gmd/kaiyou/english/co2_flux/co2_flux_data_en.html | MLR | Cluster dependent linear trends | Manually defined clusters by regions. | SST, SSS, SSH, CHL, MLD, TIME. |
| MPI-SOMFFN | 1982-2019 | Landschützer et al. (2016). (https://www.ncei.noaa.gov/data/oceans/ncei/ocads/data/0160558/MPI_SOM-FFN_v2021/) | FNN | Implicit rates learned from air CO2 | 16 clusters by self-organization map. | SST, SSS, MLD, CHL, CO2A. |
| CMEMS-FFNN | 1985-2019 | Chau et al. (2021). (https://resources.marine.copernicus.eu/?option=com_csw&view=details&product_id=MULTIOBS_GLO_BIO_CARBON_SURFACE_REP_015_008) | FNN | Implicit rates learned from air CO2 | Clustering by month with a window size of three months. | SST, SSS, SSH, MLD, CHL, CO2A, CO2C, LAT, LON. |
| CSIR-ML6 | 1982-2016 | Gregor et al. (2019) (https://www.ncei.noaa.gov/access/metadata/landing-page/bin/iso?id=gov.noaa.nodc:0206205) | GBM, FNN, SVR, and RF | Implicit rates learned from air CO2 and time | Repetition of K-mean clustering and CO2 biomes clustering | SST, dSST, SSS, MLD, CHL, dCHL, WIND, CO2A, TIME. |
| OceanSODA-ETHZ | 1982-2020 | Gregor and Gruber (2021). (https://doi.org/10.25921/m5wx-ja34) | GBM and FNN | Implicit rates learned from air CO2 | Repetition of K-mean clustering. | SST, SSS, CHL, MLD, WIND, CO2A. |



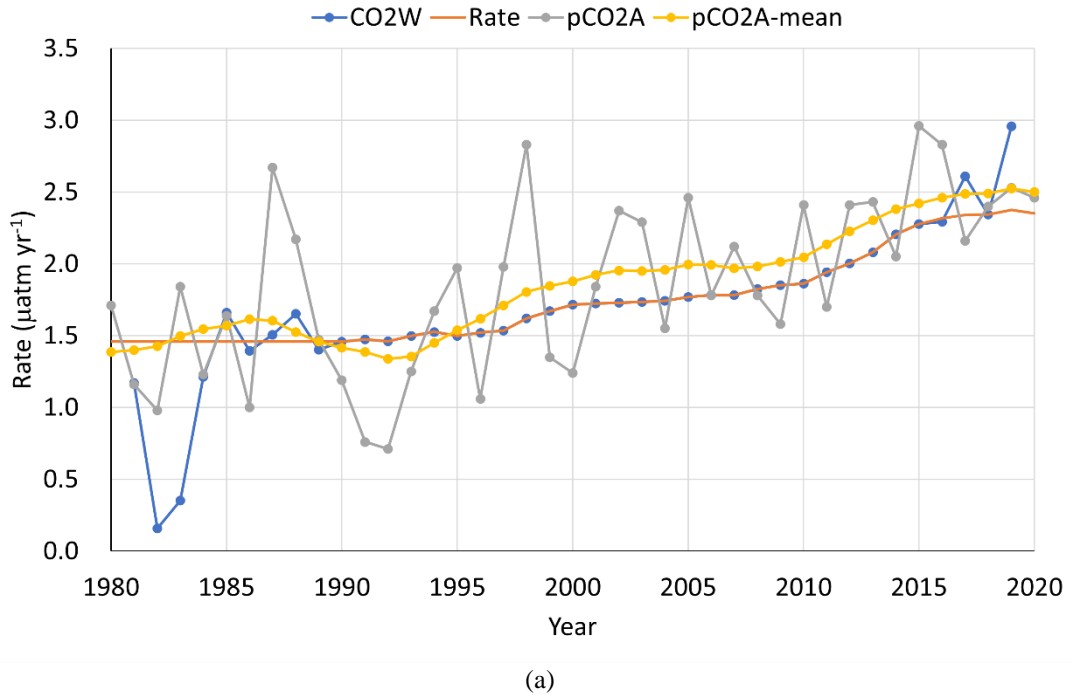

(a)

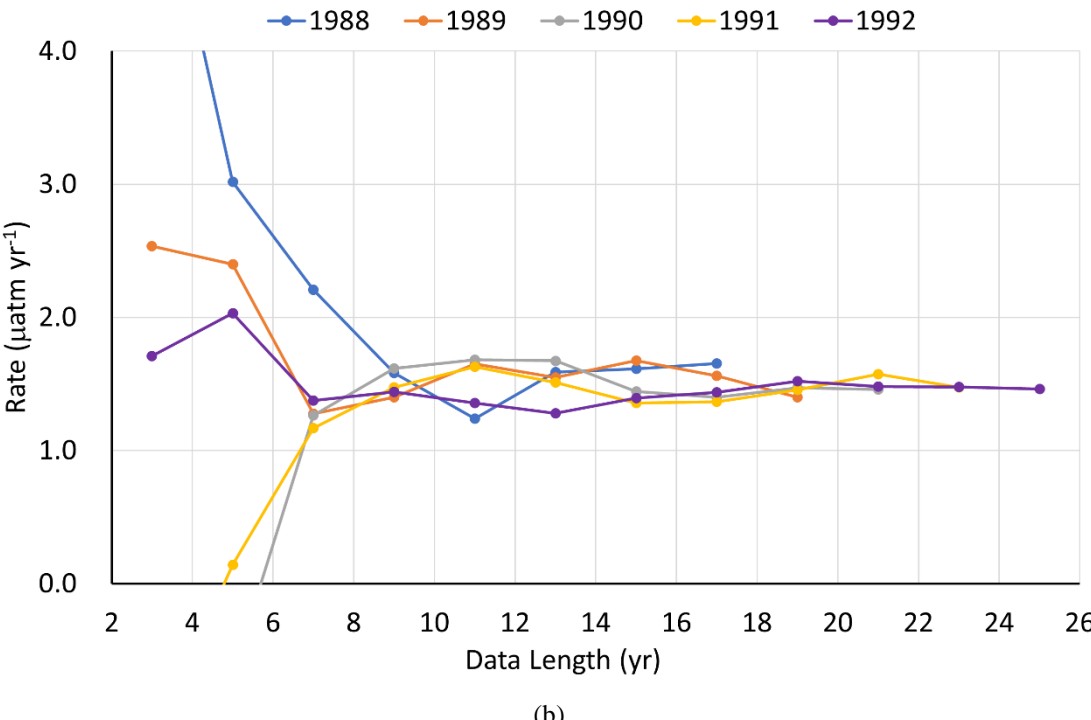

(b)




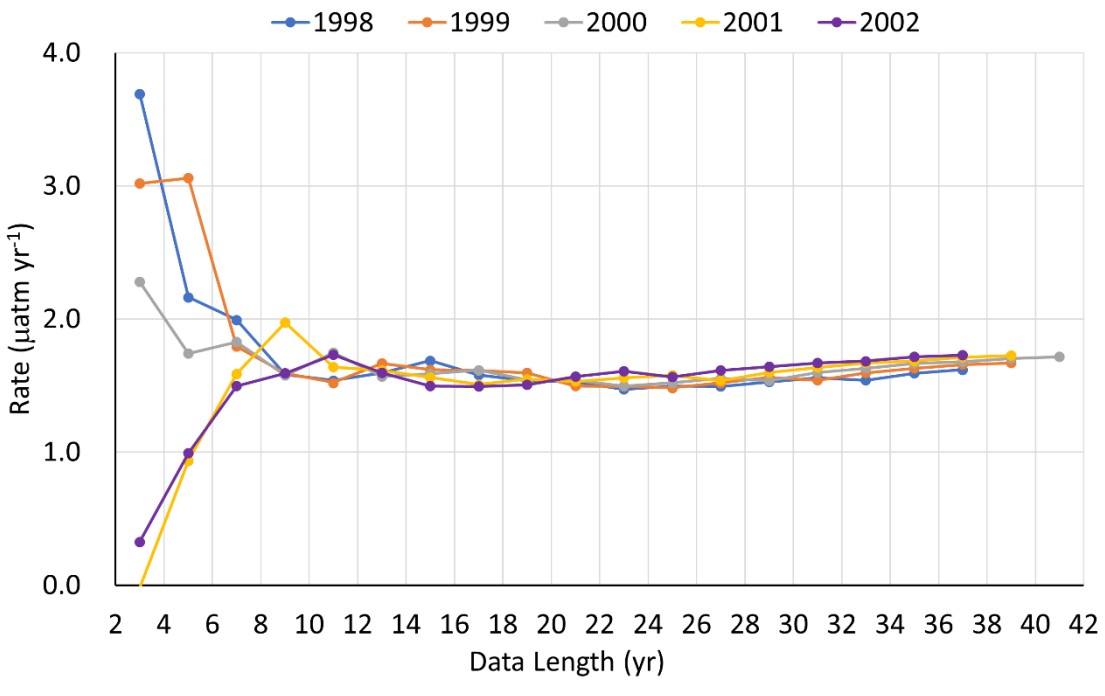

(c)

(d)



**Figure 1: Annual increase rate of CO₂. (a): the annual increate rate (blue) for a target year was estimated by using the iteration method with the longest data length around the year; the rates used for normalization (orange) are the same as those in blue except for the period when data are insufficient; the annual increase rates of atmospheric CO₂ (grey) and decadal means (yellow) are presented as references. (b)-(c) demonstrates the variations of the rates with data length and the difficulty of choosing the optimal length.**


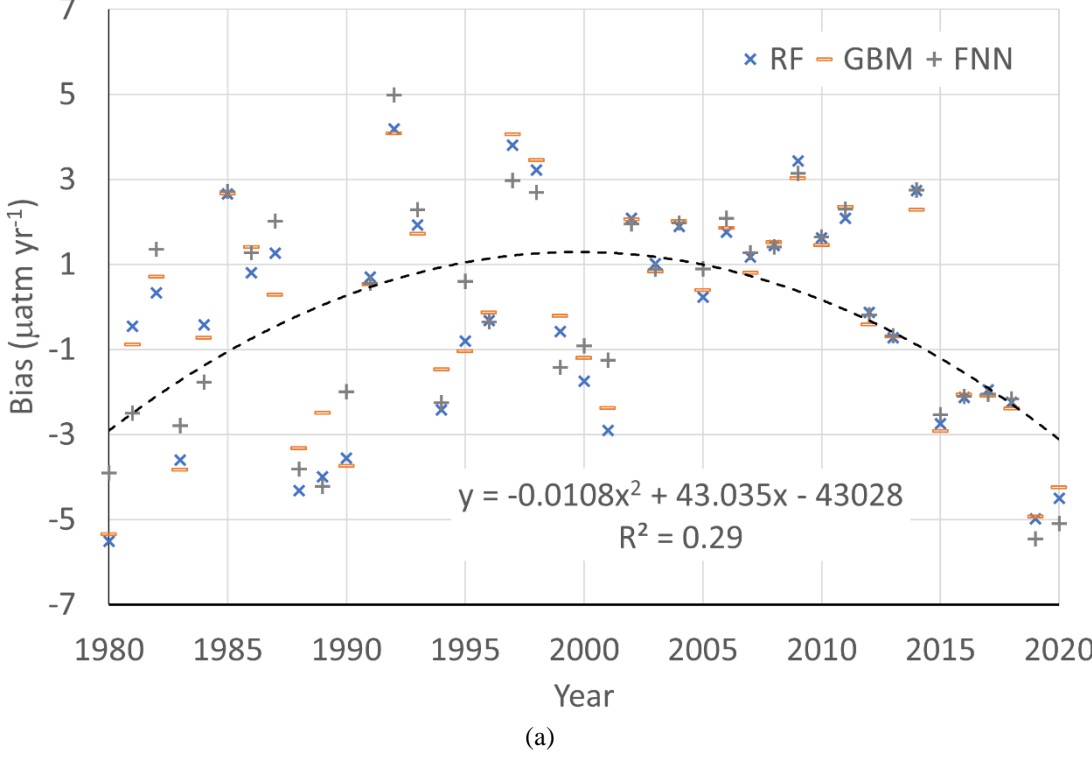

(a)



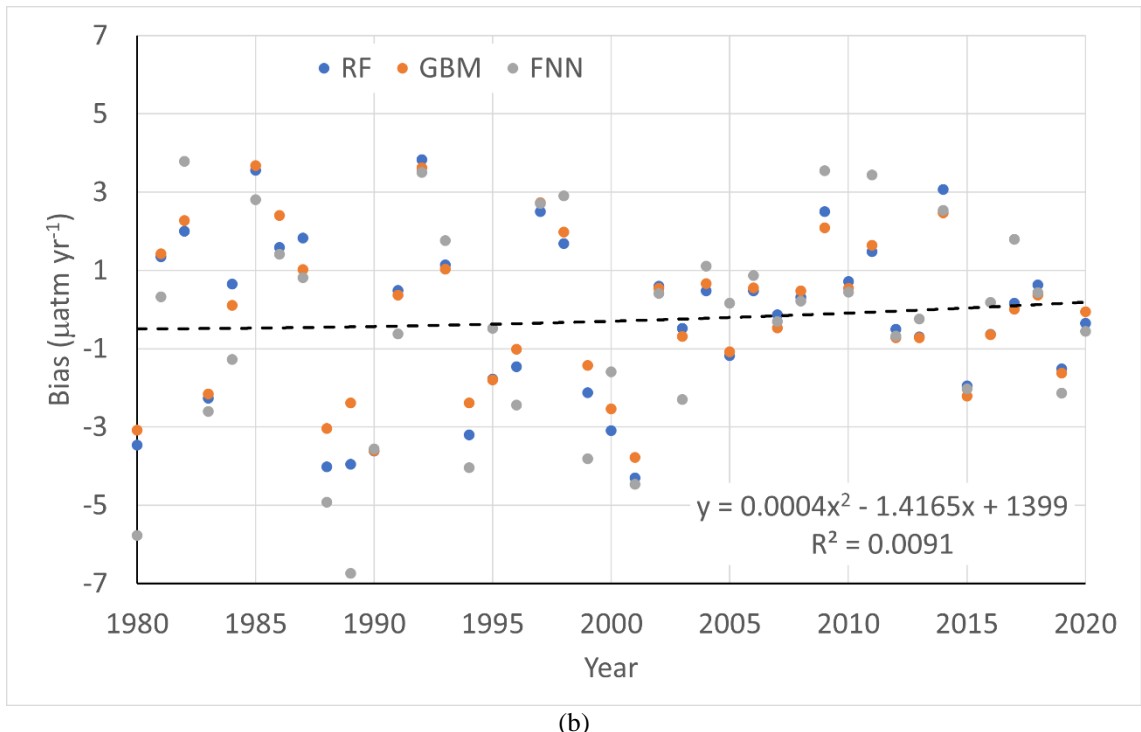

(b)

**Figure 2: Trend of LOYO biases: (a) results of applying LOYO to data normalized with a constant rate of 1.593 μatm yr⁻¹. (b) results of applying LOYO to data normalized with the rates in Table 2.**

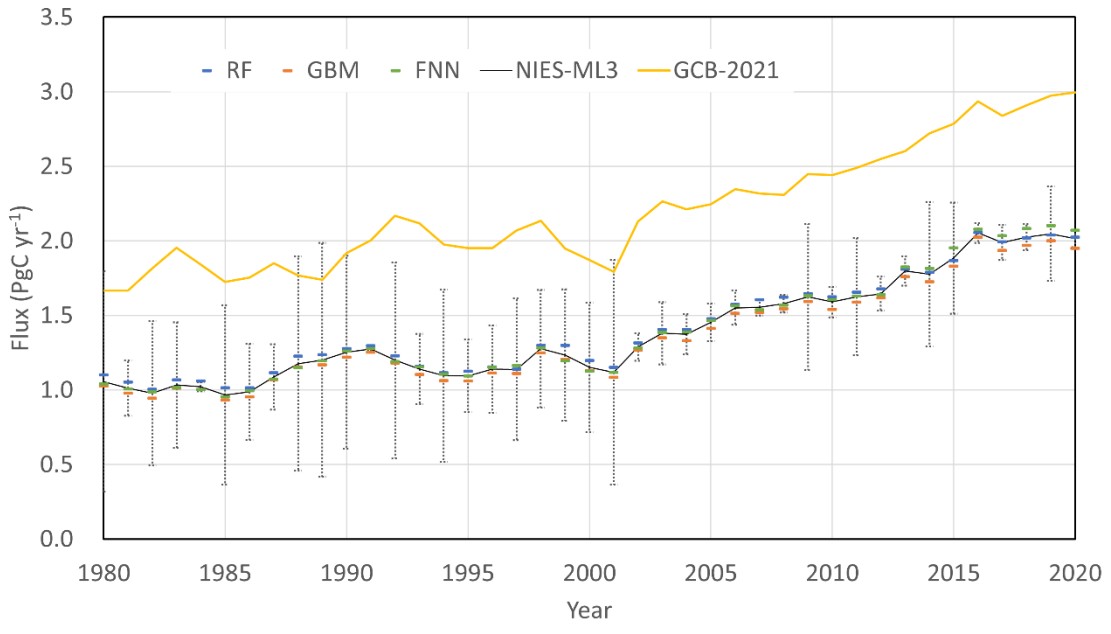

**Figure 3: Annual air-sea fluxes. The grey line (NIES-ML3) represents the mean fluxes of the random forest (blue), gradient boost**
**machine (orange), and feedforward neural network (green) models. The error bars are uncertainty estimates using the mean biases**
**of the three models in Table 2. The yellow line shows the mean fluxes of the biogeochemical models in Global Carbon Budget 2021.**
**The difference between NIES-ML3 and GBC-2021 can be attributed to riverine input and coastal areas not included in NIES-ML3.**

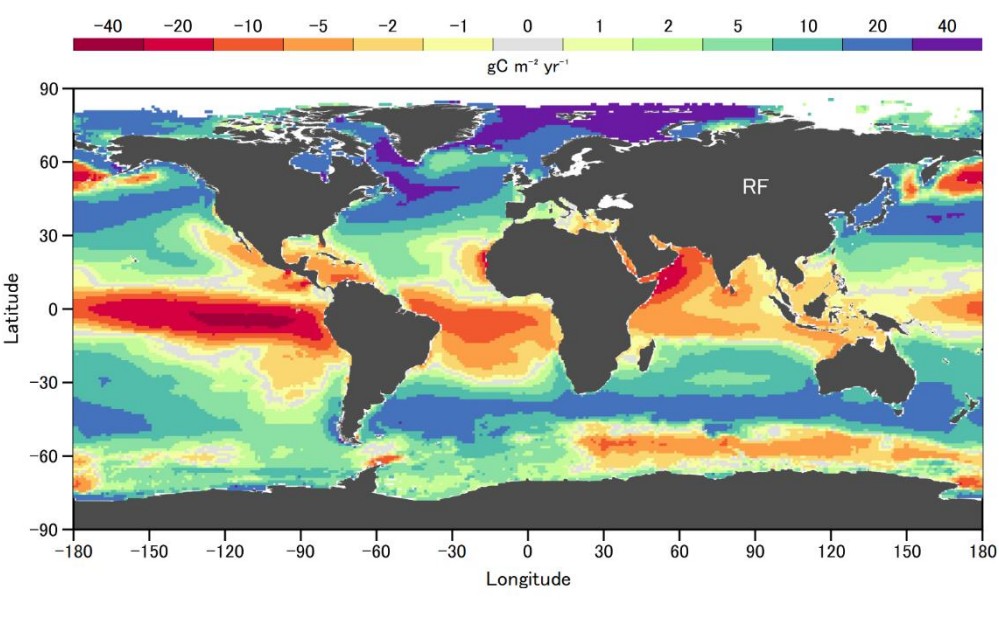

535                                                        (a)

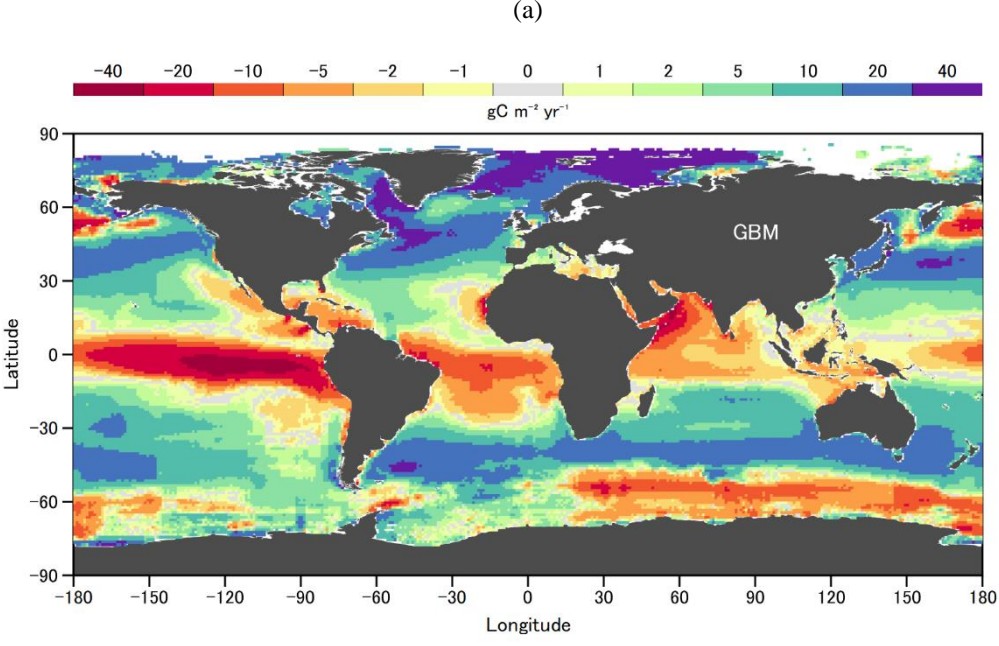



(b)

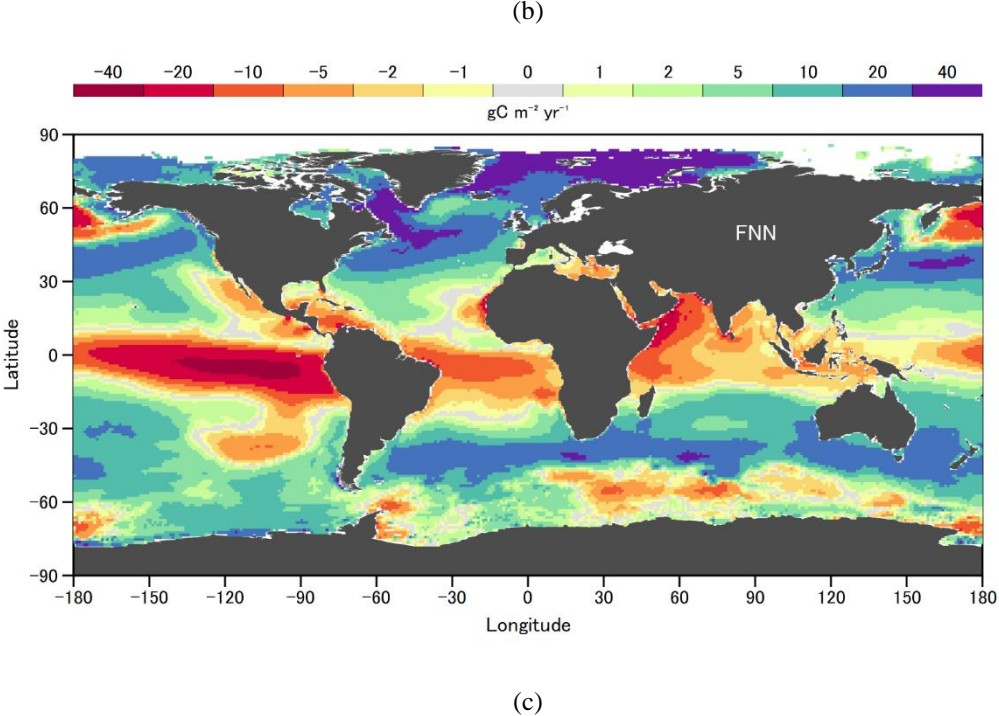

(c)


**Figure 4: Distribution of the mean annual air-sea flux in 1980-2020. The patterns agree well with those in GCB-2021. Large sinks are located above 30°N and in the 30°S-60°S zone.**



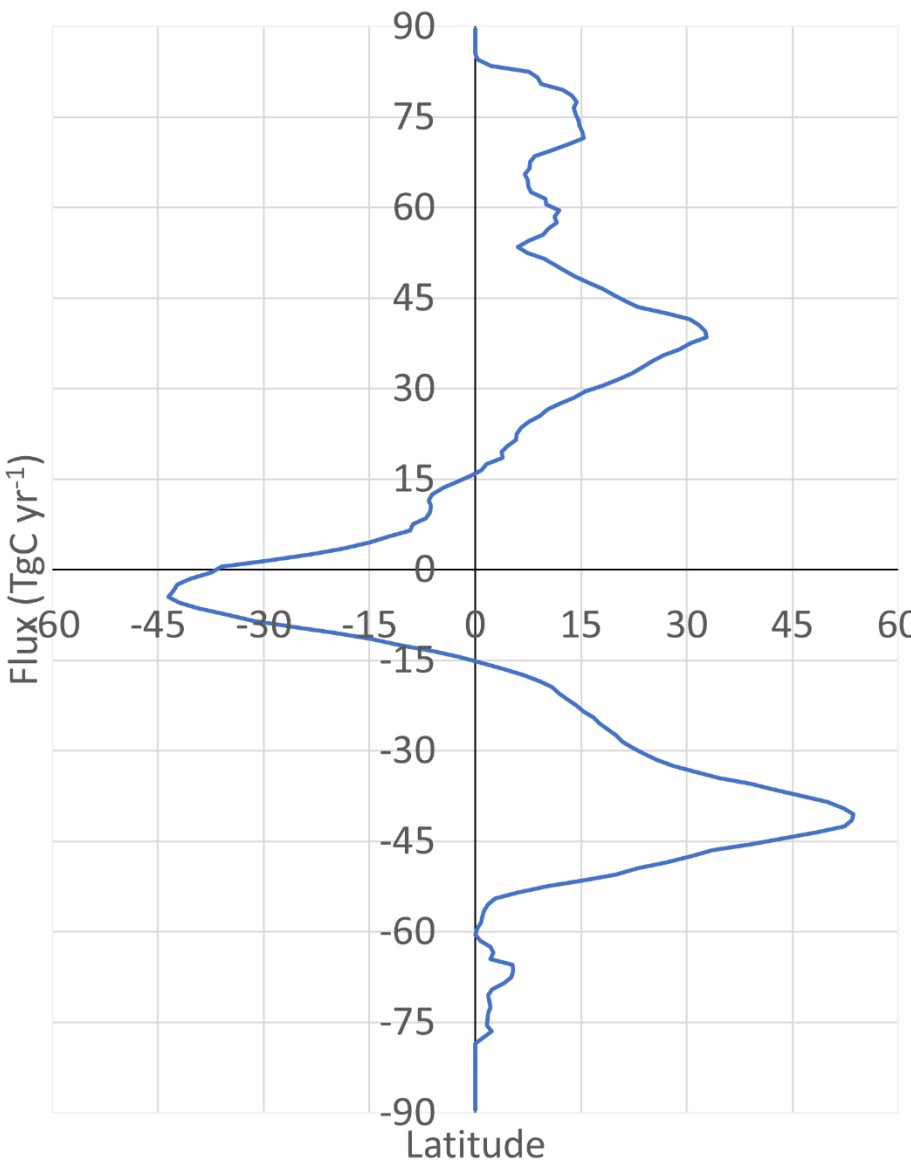

**Figure 5: Latitudinal profile of air-sea fluxes. The tropical areas were the net emitter and the oceans in mid latitude played as important sinks.**





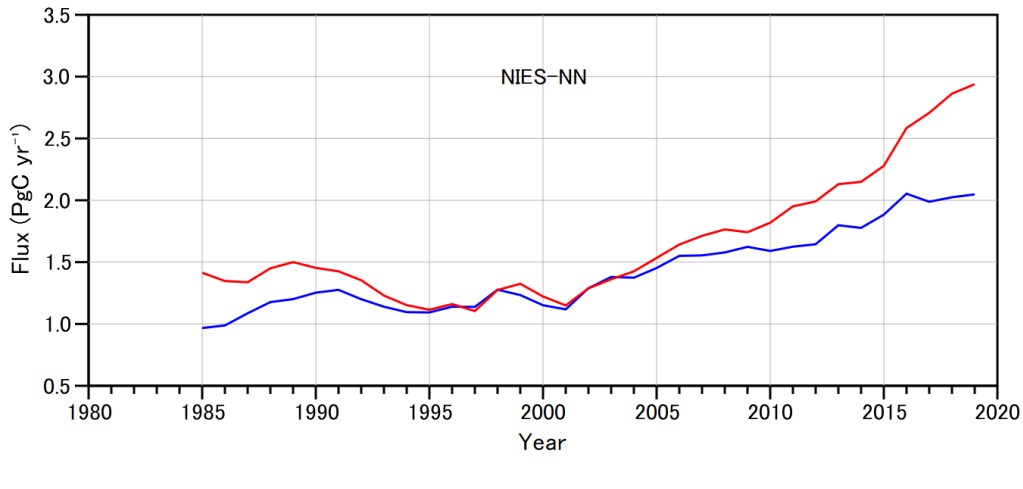

(a)

(b)






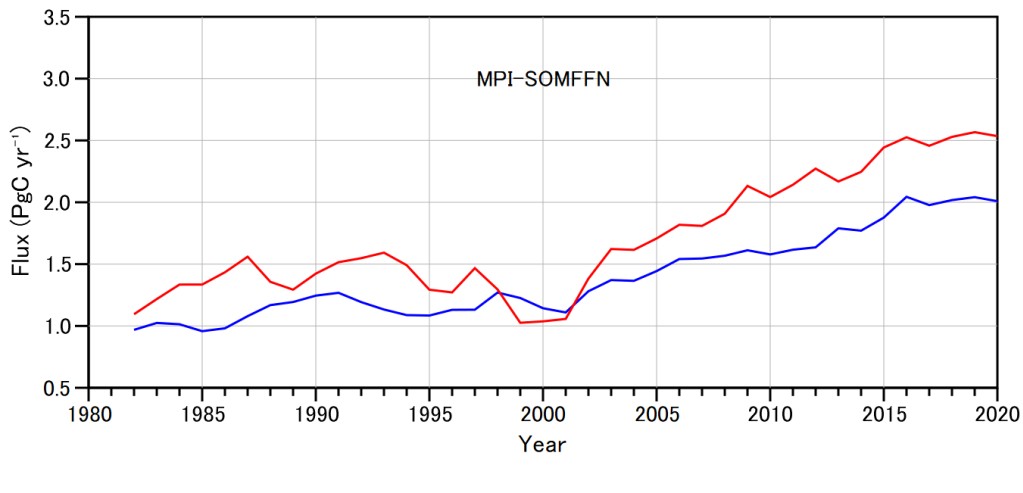

(c)

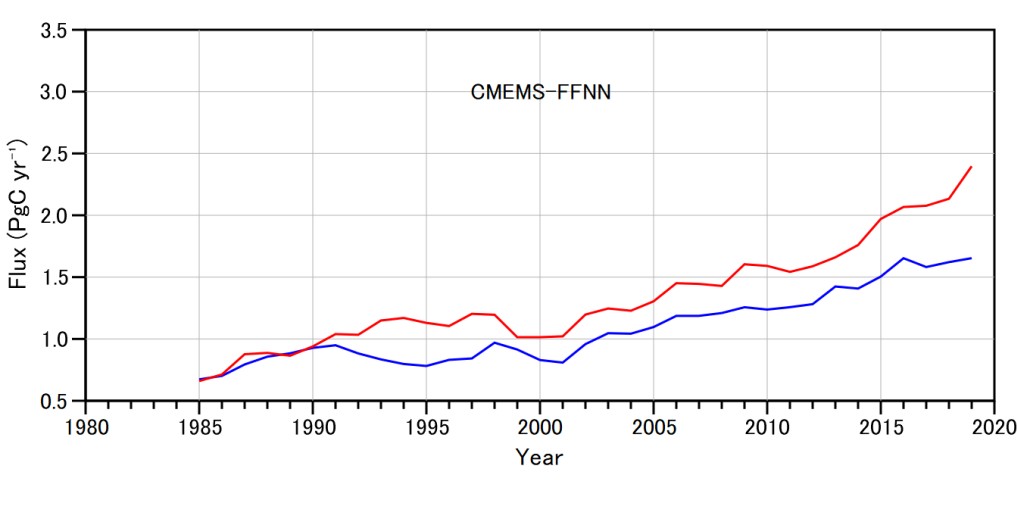


(d)

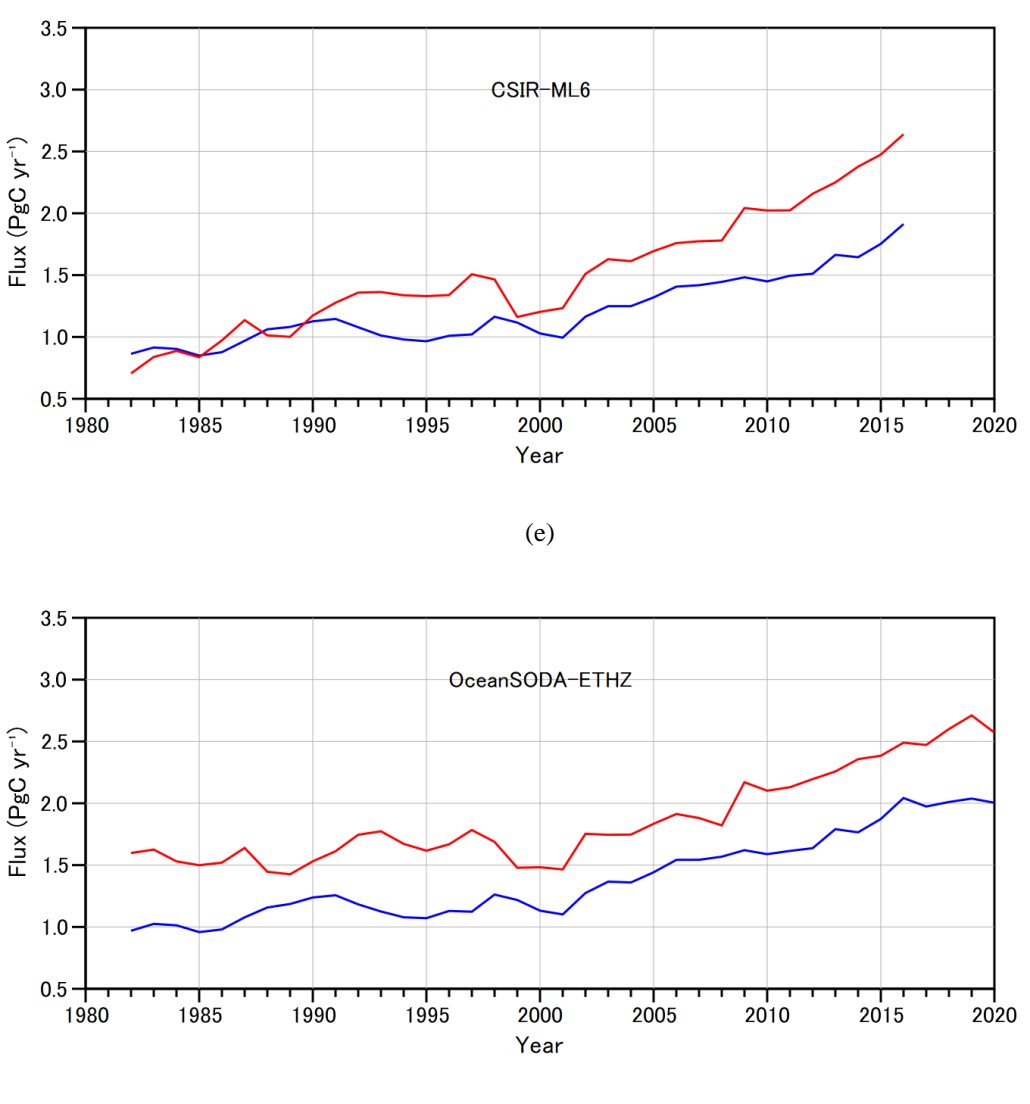

(e)

(f)

**Figure 6: Comparisons with products included in GSB-2021. NIES-ML3 and the product under comparison are present by blue and red lines respectively. The comparisons are relative because of different spatial coverage and different parameters and wind products used for flux calculation. Only those grids where both products have data were counted to calculate the global sum.**