# Peer review of "A new estimate of oceanic CO2 fluxes by machine learning reveals the impact of CO2 trends in different methods"

_Earth System Science Data, 2022_

## Referee Comment (RC1)

Zeng et al. used three machine algorithms (neural network, random forest, and gradient boosting) to estimate ocean $pCO_2$ on a 1x1 grid from 1980-2020. They trained each algorithm to learn SOCAT $fCO_2$ observations using full-coverage fields (SST, SSS, MLD, CHL, LAT, LON, YEAR) as inputs to each algorithm. The output from these algorithms were averaged to create the final product and a bulk parameterization was used to estimate flux. Their flux estimates were lower than the 6 products used in the global carbon budget 2021.

**Major comments:**
A lot of the methods used in this manuscript are poorly described, leaving the reader wondering whether best practices were used. A validation dataset separate from the training and testing datasets is typically used to determine the model architecture. The authors state on line 110 that data post-1980 was used to train the tree algorithms. There was no mention of whether an independent test-set was withheld or how the architecture of each algorithm was selected. For example, how did the authors decide on the number of layers, number of nodes, optimization algorithm, and weight initialization for the neural network? For the tree-based algorithms (random forest, gradient boosting) how did the authors decide on the number of trees and number of samples used to construct each tree?

When calculating atmospheric $pCO_2$ the authors need to take into account the water vapor correction. The authors use the marine boundary reference product which uses dry-air mole fraction of $CO_2$. Because of this, a correction for the water vapor that was removed needs to be taken into account, see Dickson et al. (2007). This correction is small can become important in the delta-$pCO_2$ when calculating flux.

On line 124 the authors state the gas-transfer velocity and solubility were calculated following Wanninkhof (2014). I would have liked to see more discussion on this since the datasets used to calculated these terms were not clearly stated. The gas-transfer velocity depends on wind speed. However, if you estimate $CO_2$ flux with a monthly resolution then the wind speed variance needs to be taken into account, see Wanninkhof (1992 and 2014). I am concerned the variance in the wind speed was not taken into account. The authors state they used monthly wind from ERA5 to calculate atmospheric $pCO_2$, but there was no mention whether the monthly variance was used to calculate fluxes.

The evaluation of the product needs significant improvements. It is standard practice to test machine learning algorithms against a withheld test-set. This was either not done or not mentioned in the manuscript. I suspect this was not done since the authors state post-1980 observations were used to train the algorithms. There was no comparison to any independent datasets. The six products used in the GCB2021 compare to estimates of $pCO_2$ from HOT and BATS time series and GLODAP, just to name a few. Even though $pCO_2$ at HOT, BATS, and GLODAP is not directly measured, but inferred from the carbonate cycle, I strongly suggest this comparison be done since SOCAT observations are sparse, which the authors mention.

Instead, the authors compare their results against the 6 products in the GCB2021. This will tell you how well the algorithm compares to other products, but is not the same as comparing to

independent observations. Even this comparison to the other products has some flaws. Instead of comparing the pCO2 output the authors compare the flux outputs. They compare the flux from their product to the flux from the others, taking into account the differences in spatial and temporal coverage between the products. My issue with this is they do not re-calculate the flux for each product using a consistent approach. Instead, they compare to the flux calculated by each product's practitioner. The authors do acknowledge that the different wind products used by each practitioner can influence this comparison. I suggest the authors refer to Fay et al. (2021), which addresses this issue. The authors should re-calculate the flux from the products pCO2 using a consistent approach to alleviate this issue. I would also suggest comparing their flux against anthropogenic flux estimates from Denman et al. (2007) and Gruber et al. (2019).

**Minor comments:**
The authors learn SOCAT fCO2 and then convert to convert to pCO2 after using the method of Weiss (1974). The six pCO2 products used in the GCB2021 that the authors compare to do this conversion prior to training. I am curious if this has an impact on the results and which datasets were used in this conversion. For instance, the conversion requires SST and there was no mention of which dataset was used in this conversion.

**References**
Denman, K. L., Brasseur, G. P., Chidthaisong, A., Ciais, P., Cox, P. M., Dickinson, R. E., & Steffen, W. (2007). Couplings between changes in the climate system and biogeochemistry. In Climate change 2007: The physical science basis (pp. 499– 588). Cambridge University Press.

Dickson, A. G., Sabine, C. L., & Christian, J. R. (2007). Guide to best practices for ocean CO2 measurements. In PICES Special Publication 3 (pp. 191).

Fay, A. R., et al. (2021). SeaFlux: harmonization of air–sea CO 2 fluxes from surface pCO2 data products using a standardized approach. Earth System Science Data, 13(10), 4693-4710.

Gruber, N., Clement, D., Carter, B. R., Feely, R. A., Van Heuven, S., Hoppema, M., et al. (2019). The oceanic sink for anthropogenic CO2 from 1994 to 2007. Science, 363(6432), 1193– 1199.

Wanninkhof, R. (1992). Relationship between wind speed and gas exchange over the ocean. Journal of Geophysical Research, 97(C5), 7373– 7382.

Wanninkhof, R. (2014). Relationship between wind speed and gas exchange over the ocean revisited. Limnology and Oceanography: Methods, 12(6), 351– 362.

---

## Author Comment (AC2)

The reader has made valuable comments. We have recalculated fluxes of all products used in the comparison, compared fluxes of NIES-ML2 obtained by CCMP and ERA5 wind, and revised the manuscript to include the changes. The followings are out point-to-point response to the reader's comments.

1. Our validations were discussed in the "Model Performance" section. Maybe because of the title, the reader thought we did set data aside for validation. Now we changed the title to "Model Validation". There are different ways to validate a machine learning model. A common method is to randomly divide a data set into two parts, one for training and another for validation. We believe this is not a good practice as random sampling would make the two parts having the same distribution and a validation won't fail unless no relationship exists between the target and the predicters. We believe it would better that the sampling domain of the testing data differs from that of the training data. We used a so-called leave-one-year-out method, that is that for each year from 1980 to 2020, one year's data was set aside for validation and others for training. Thus 41 validations were done. The results were summarized in Table 2 of the manuscript.

2. Regarding model configuration. It is a complicated issue and there is no universal method to obtain the "best configuration" for a given problem. We selected the configuration parameters for RF based on our experience with using RF for global forest GPP reconstruction and the lesson we learnt from ocean CO2 reconstruction in the past. Few data are available in the southern oceans in some months (we added an example the supplement material). An overfitting would result in hot spots in the areas. So we raised the default number of trees and end-node leaves to prevent the problem. As GBM is also a tree-based model, we opted to use the same parameters. As for the FNN, we wrote the model code used in Zeng et al. (2014) and did extensive test at that time. The full-batch method of the code is very slow with the data size of this study. It would take a few months if we used the code to do the same calculations (a lot of iterations are involved for a rate extraction of many years). We switch to use python's MLPRegressor. Its mini-batch method training is much faster. We did a few tests by comparing its results with our old program and figuration seems working well.

3. We did water vapor correction. The expressions on the matter in our manuscript are misleading. (We revised them.) The CO2 from NOAA's Marine Boundary Layer

Reference is mole fraction (xCO2). We converted xCO2 to pCO2 by pCO2=xCO2*($P_s$-$P_{h2o}$). The vapor pressure of seawater $P_{h2o}$ was calculated by the method of Weiss and Price (1980).

4. We recalculated out fluxes with $\alpha$=0.271 for ERA5 wind and $\alpha$=0.257 for CCMP wind. We also recalculated fluxes of the products used for comparison with the same method and adjusted the fluxes as if the products have the same spatial coverage of NIES-ML3 so that they can be put together in one figure for comparison.

5. As the reader pointed out, pCO2 of HOT, BATS, and GLODAP are not directly measured. We don't think it is logical to use them for validation. It would be difficult to judge that a disagreement is resulted from the method or from the systematic difference between SOCAT and GLODAP. It would be better to merge the two datasets for CO2 reconstruction. Even doing so won't solve the data scarcity program in southern oceans in boreal summer (refer to supplement Fig. 1d and https://essd.copernicus.org/articles/12/3653/2020/#section5&gid=1&pid=1). Also, using data from a few sites cannot draw any statistically sound conclusion. In our validations, when a year's data was set aside for testing, the bias is far from zero even though thousands of data point were included. But the overall bias of 41 validations is negligible.

6. The reader also suggests comparing our results with those of Denman et al. (2007) and Gruber et al. (2019). Unfortunate their time series are short. Gruber's publication is quite new, but the data used is more than 10 years older. Our study emphasizes how the annual increase rates of CO2 used or embedded in different methods could affect the long-term variation of CO2 flux. That's the reason we didn't use SeaFlux for comparison in the first place because the product only include estimate after 1990.